# Isolation of a Divergent Strain of Bovine Parainfluenza Virus Type 3 (BPIV3) Infecting Cattle in China

**DOI:** 10.3390/v11060489

**Published:** 2019-05-29

**Authors:** Élcio Leal, Cun Liu, Zhanzhong Zhao, Yong Deng, Fabiola Villanova, Lin Liang, Jinxiang Li, Shangjin Cui

**Affiliations:** 1Institute of Animal Sciences, Chinese Academy of Agricultural Sciences, Beijing 100193, China; liucun89@163.com (C.L.); zhaozhanzhong@caas.cn (Z.Z.); liang-lianglin@caas.cn (L.L.); 2Federal University of Pará, Belém 66075-000, Brazil; fvillanova@gmail.com; 3Beijing Observation Station for Veterinary Drug and Veterinary Biotechnology, Ministry of Agriculture, Beijing 100193, China; 4China Institute of Veterinary Drugs Control, Beijing 100083, China; yftk_007@163.com

**Keywords:** bovine parainfluenza virus, vaccine, phylogenetic analysis, MDBK cells, cattle, biotechnology

## Abstract

Bovine parainfluenza virus type 3 (BPIV3) is one of the most important known viral respiratory pathogens of both young and adult cattle. It is also named “heat stress in transport”, causing morbidity and mass death. New variants of BPIV3 have been detected or isolated in China since 2008. Here, we isolate one BPIV3 strain (named BPIV3 BJ) in Madin-Darby bovine kidney (MDBK) cells from nasal samples collected in China. Phylogenetic analysis showed that our isolate is related to BPIV3 of the genotype A. The comparison of BPIV3-BJ and the reference Chinese isolate NM09 showed that these strains are highly divergent. We found many differences in the amino acid composition in the nucleocapsid (NP) protein among these genotype A strains. Since the NP protein has been implicated in immunization studies, our BPIV3 isolate will be useful for the development of immune assays and vaccine studies. The diversity of BPIV3 lineages that we found in China indicated ongoing evolution for immune escape. Our study highlights the importance of genetic surveillance for determining the effect of BPIV3 variability on pathogen evolution and population-scale immunity.

## 1. Introduction

Bovine respiratory disease complex (BRDC), a multi-factorial disease, is an economically important health problem of cattle worldwide. The disease is commonly referred to as “Shipping fever” and causes an increase in morbidity mortality rates [1]. The multiple factors that cause BRDC include stress, infectious agents, immunity, and housing conditions. The infectious agents associated with BRDC include viruses, bacteria, and mycoplasmas [2,3]. While most acute infections with uncomplicated infectious agents are sub-clinical, they can cause respiratory disease characterized by a cough, fever, and nasal discharge [4]. Mixed infections with two or more infectious agents are thought to contribute to BRDC [5]. The primary viral infectious pathogens that cause BRDC are bovine parainfluenza virus 3 (BPIV3), bovine respiratory syncytial virus (BRSV), bovine viral diarrhea virus (BVDV), bovine alphaherpesvirus 1 (BHV-1), bovine coronavirus (BCV), and so forth [6,7].

Bovine parainfluenza virus type 3 (BPIV3) was one of the most important viruses associated with BRDC in cattle [3,5]. It was first isolated in 1959 and first identified in cases of BRDC [3]. BPIV3 is an enveloped, non-segmented negative-strand RNA virus within the genus Respirovirus. BPIV3 induces respiratory tract damage and immunosuppression. More severe secondary bacterial and mycoplasma infections are caused in susceptible animals in instances of high stress, such as transportation and feedlot situations [4].

Up to now, based on phylogenetic analysis, BPIV3 has been divided into three genotypes: Genotype A, genotype B, and genotype C [6,8,9,10,11,12,13,14,15,16]. Multiple BPIV3 genotype A strains have been isolated in USA, China, Argentina, and Japan [8,10,12,16]. Genotype B was initially identified in Australia [6,8,9,10,11,12,13,14,15,16]. Isolation of BPIV3 genotype C, first identified in China, has also been conducted in South Korea, Japan, Argentina, and USA [5,8,10,11,15,16]. A high seropositivity rate for BPIV3 in dairy cattle indicated that a high level of BPIV3 infections occurs. Many efforts have been made focusing on the prevention and control of BRDC in order to reduce production losses in the livestock industry [17,18].

Here, we describe the cell culture isolation and genomic sequencing of a BPIV3 genotype A strain isolated from cattle in China. Although BPIV3 is endemic in cattle, little is known about the pathogenesis of this virus and information regarding antigenic variation owing to the genetic variability is rare [5]. The phylogenetic comparison of our isolated strain with strains previously characterized in China indicated the presence of divergent strains of genotype A circulating in the country. The diversity of BPIV3 in China seems to mirror the diversity of this virus, which is observed in the USA [8,11,15]. In addition, the full characterization of our BPIV3 genotype A strain will lend support to molecular diagnoses and to future studies aimed at developing an efficient vaccine against multiple viral lineages.

## 2. Materials and Methods

### 2.1. Sample Treatment and PCR Detection

Ten nasal swabs from cattle with a slight cough and nasal discharge in an auction market were collected from Shandong Province, China, in 2010. Nasal swabs were placed in virus collection tubes (Yocon Bio. Co. Ltd., Beijing, China). RNA extraction was conducted following the manufacturer’s instructions accompanying the Viral RNA Rapid Extraction Kit (Aidlab Biotechnologies Co., Ltd., Beijing, China). Potential infectious pathogens of BRDC, including BPIV3, BVDV, BHV-1, and BRSV, were detected by PCR. Nasal swabs were placed in virus collection tubes (Yocon Bio. Co. Ltd., Beijing, China). RNA extraction was conducted following the manufacturer’s instructions accompanying the Viral RNA Rapid Extraction Kit (Aidlab Biotechnologies Co., Ltd., Beijing, China). Potential infectious pathogens of BRDC, including BPIV3, BVDV, BHV-1, and BRSV, were detected by PCR. The conditions of PCRs for the detection of BVDV, BPIV3, and BRSV were as follows: Pre-denaturation at 95 °C for 3 min; denaturation at 95 °C for 30 s, annealing at 55 °C for 30 s, extension at 72 °C for 40 s, and 35 cycles for this stage; extension at 72 °C for 10 min; and storing at 4 °C for forever. The conditions of PCRs for the detection of BHV-1 were as follows: Pre-denaturation at 95 °C for 5 min; denaturation at 95 °C for 50 s, annealing at 60 °C for 40s, extension at 72 °C for 30 s, and 35 cycles for this stage; extension at 72 °C for 10 min; and storing at 4 °C for forever. Nasal samples, detected as BPIV3-positive, were used for virus isolation on Madin-Darby bovine kidney (MDBK) cells. Primers for PCR detection are shown in Appendix A.

### 2.2. Cell Cultivation and Virus Isolation

Bovine kidney cells (MDBK/NBL-1; ATCC^®^ CCL-22™) were cultured at 37 °C with 5% CO_2_, in DMEM (Fisher Scientific, Loughborough, UK) supplemented with 8% horse serum. Virus isolation and determination of the median of tissue culture infective dose were performed on MDBK cells. Virus isolation was performed as follows: The three nasal swabs, positive for BPIV3, were filtrated through a 0.22 μm filter (Millipore, Milford, MA, USA) and inoculated to a monolayer culture of MDBK cells cultured in Dulbecco’s modified eagle medium (DMEM, Fisher Scientific, Loughborough, UK) supplemented with 8% horse serum (Fisher Scientific, Loughborough, UK). MDBK cells were maintained at 37 °C in an atmosphere of 5% CO_2_. The cytopathic effect (CPE) was examined daily. The storage solution was exposed to a ten-fold dilution and filtered with a 0.22 μm filter. Then, the filtrate was inoculated into MDBK cells for 1 h. Finally, the culture medium was replaced with DMEM containing 2% horse serum. MDBK cells inoculated with filtrate were cultured in an incubator continually for 72 h. The propagation of the virus was performed three times.

### 2.3. Cloning and Whole Genome Sequencing

The third-generation virus sub-cultured in MDBK was used for complete genome amplification. Twelve pairs of primers (described in Appendix A) were used for the whole genome amplification. PCR products were purified, cloned into pMD18-T, and sequenced. Sequences data were compiled to generate the complete genome sequence of BPIV3. Sequences were assembled using SeqMan (DNASTAR, Madison, WI, USA).

### 2.4. Sequence Analysis

The complete genome sequences of BPIV3 available in GenBank were used for genetic and phylogenetic analysis. BLASTn was initially used to identify viral sequences through their sequence similarity to annotated viral genomes in GenBank. Based on the best hits of blastx searches, the following 24 complete genomes were choose for the next analyses: Genbank numbers: KU198929; KT071671; JX969001; KJ647287; KJ647285; LC000638; LC040886; EU277658; KJ647284; KJ647286; KP764763; KJ647289; JQ063064; KP757872; D84095; AB770485; AB770484; KJ647288; EU439428; AF178654; EU439429; AF178655; and HQ530153. These genomes were then aligned using Clustal X software [19]. Subsequently, a phylogenetic tree was constructed by the Maximum Composite Likelihood (MCL) approach assuming the Hasegawa-Kishino-Yano model plus a discrete Gamma distribution (with five categories and an estimated alpha parameter = 3.1840) and the rate of invariable sites of 54.73%. All phylogenetic analyses and tree editions were conducted using MEGAX software [20].

## 3. Results and Discussion

BPIV-3 can vary considerably, ranging from asymptomatic infections to severe respiratory illness. In the implicated infection cases, mild clinical signs were characterized by coughing, fever, and nasal discharge. Mixed infections with two or more infectious agents can cause more severe clinical presentation and production losses, due to the immunosuppression and severe bronchopneumonia from secondary bacterial infections. To investigate the potential infectious pathogens causing coughing and nasal discharge, BVDV, BPIV3, BHV-1, and BRSV were detected by PCR. The results of PCR are summarized in Figure 1.

Three nasal swab samples were positive for BPIV3. These three samples positive for BPIV3 were used for virus isolation. The samples were filtrated through a 0.22 μm filter and inoculated to monolayer culture MDBK cells. The cells were cultivated three times with five-day intervals. Cell cultures (third passage) were frozen and thawed three times before inoculating MDBK cells. MDBK cells inoculated with the virus produced typical cytopathic effects characterized by rounding, shrinkage, and cracking off (Figure 2). The successfully isolated BPIV3 stain was named BPIV3-BJ. The median of viral tissue culture infective dose (TCID50) of the third passage was determined in MDBK cells. The titer of BPIV3-BJ was up to 10–9.5/0.1 mL. The high titer of BPIV3-BJ on MDBK indicated that the virus has a good growth performance in MDBK cells.

The whole genome of BPIV3-BJ was assembled using SeqMan. The genomic annotation was performed according to the blastx search. The complete genomic length of BPIV3-BJ was 15,480 bp with 36.1% GC content. The percentage of nucleotide sequence similarities was assessed with these BPIV3 strains for which complete genomic sequences were available in GenBank. BPIV3-BJ showed a low nucleotide similarity to BPIV3 strains located in genotype B (83%–83.6%) and genotype C (81.7%–82.1%). It shared a higher nucleotide identity with isolates located in genotype A. In total, 99% of nucleotide identity was observed between BPIV3-BJ and 910N (Genbank number: D84095). The virus isolated in this study showed a lower nucleotide identity with Chinese BPIV3 strains. 

Phylogenetic analysis showed that BPIV3-BJ (opened diamond in Figure 3) is located within the clade composed of BPIV3 isolates of the genotype A. This clade (clade A) includes the strain 910N isolated in Japan in 1987 (filled diamond in Figure 3) and the divergent strain NM09 (Genbank number: JQ063064) recently isolated in China. The tree also indicates the position of the isolate SD0835 (opened circle in Figure 1) in the cluster of genotype C. It is important to mention that all clades of genotype A include the BPIV3 strain from the USA and some of these American strains were isolated before the 1990s. In addition, regardless of the date, the American strains were isolated, they are located at the base of the clades. These facts suggest that the global dissemination of BPIV3 of genotype A is continuously spread from American strains to other countries.

We also estimated genetic distances within each clade (values within rectangles in Figure 3) to show that the amino acid diversity among isolates of clade A (0.068 ± 0.002) is higher than that of clade B (0.043 ± 0.001) and clade C (0.017 ± 0.001). 

The differences in the nucleocapsid (NP) protein among some strains are summarized in Table 1. Since the NP protein has been implicated in immunization studies, it is important to determine the extent of differences between BPIV3-BJ and other strains. The comparison of BPIV3-BJ and the Chinese isolate NM09 showed that these strains are highly divergent. Although there are few BPIV3 complete genomes available, the genetic divergence of strains of genotype A suggests that this genotype is continuously disseminated in distinct countries. Our study showed that the diversity of genotype A strains in China was likely affected by incoming strains from the USA. Because China imports large amounts of beef resources from the USA and the European Union, it is quite reasonable that this trade has impacted the spread of BPIV3 of genotype A to the region. Furthermore, there are also some levels of imported or smuggled live cattle from abroad and this will certainly have had a direct impact on the maintenance of distinct BPIV3 strains in China. To better address the dynamics of BPIV3 in China, and the pathways and impact of animal trade, a very extensive survey on local herds and shipments coming from other countries should be conducted.

In summary, we have described the isolation and genomic sequencing of BPIV3 genotype A strains isolated from cattle in China. Our isolate is highly divergent from the previously described isolate from China, NM09. The BPIV3 genotype A infection has been infecting livestock animals in China and causing enormous economic loss to the cattle industry. For this reason, the isolation and genomic characterization of our divergent BPIV3 genotype A strain will help the development of molecular diagnostic tools and vaccine studies.

## Figures and Tables

**Figure 1 viruses-11-00489-f001:**
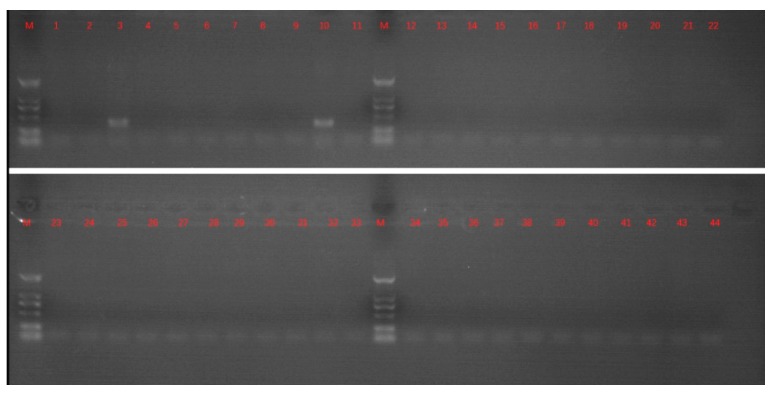
PCR detection of bovine viral diarrhea virus (BVDV), bovine parainfluenza virus type 3 (BPIV3), bovine alphaherpesvirus 1 (BHV-1), and bovine respiratory syncytial virus (BRSV) by PCR. Lane M was a 2000 bp DNA marker (DNA marker shown from top to bottom as 2000 bp, 1000 bp, 750 bp, 500 bp, 250 bp, and 100 bp); lanes 1, 12, 23, and 34 were negative controls; lanes 2–11 were ten nasal samples for BPIV3 detection, respectively; lanes 13–22 were ten nasal samples for BVDV detection, respectively; lanes 24–33 were ten nasal samples for BRSV detection, respectively; lanes 35–44 were ten nasal samples for BHV-1 detection, respectively.

**Figure 2 viruses-11-00489-f002:**
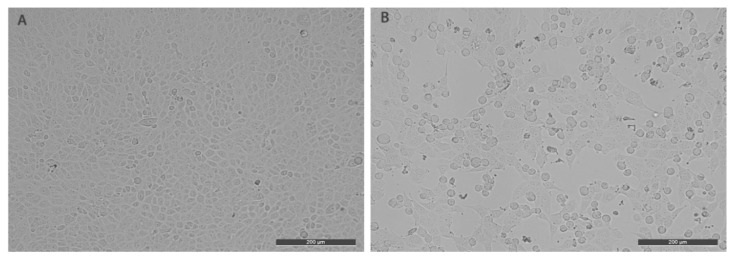
The cytopathic effect (CPE) of BPIV3-BJ in Madin-Darby bovine kidney (MDBK) cells: (**A**). The controlled MDBK cells; (**B**). the MDBK cells infected by BPIV3-BJ.

**Figure 3 viruses-11-00489-f003:**
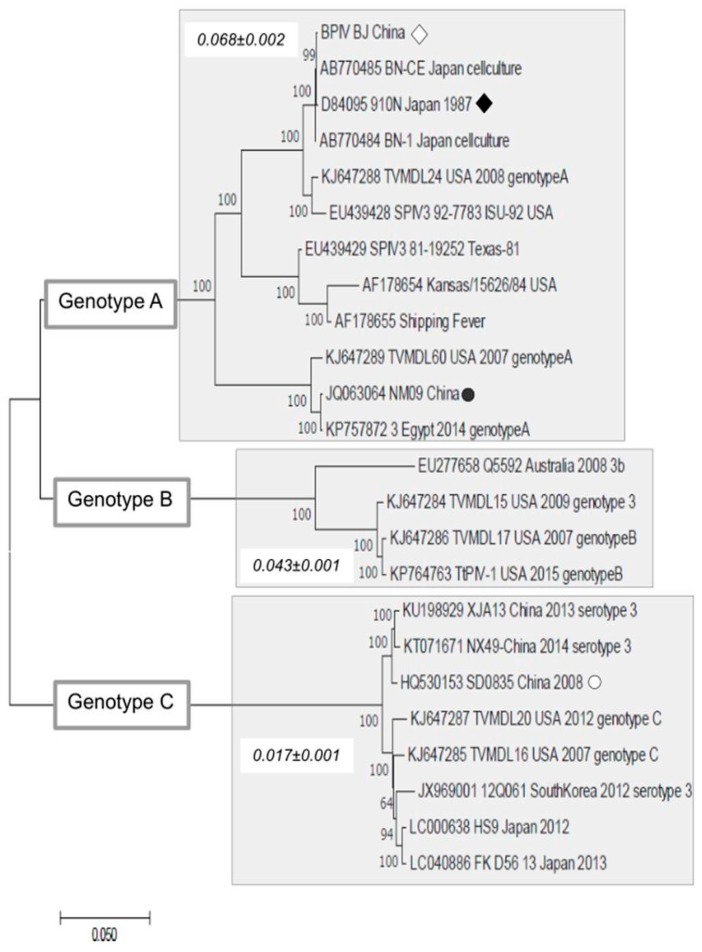
Phylogenetic trees constructing complete genomes of bovine parainfluenza virus. Phylogenetic tree constructed using 24 genomic sequences of BPIV3. The tree was inferred using a maximum composite likelihood (MCL) approach assuming the HKY model plus a discrete Gamma correction and the rate of invariable sites. Values above branches indicate bootstrap support after 500 replications. The BIPV3 strain in this work is indicated by an open rectangle in the tree. Numbers within each gray rectangle are the genetic distances and standard errors of strains of each genotype (i.e., genotype A, B, and C). Phylogenetic analyses and estimation of genetic distances were conducted using MEGAX software.

**Table 1 viruses-11-00489-t001:** Amino acid differences of the nucleocapsid (NP) protein among BPIV strains.

Protein Name (Position)	BPIV-BJ and D84095 (Position)	BPIV-BJ and JQ063064 (Position)	BPIV-BJ and HQ530153 (Position)
NP (111-1658)	none	R(556)G; V(628)I; N(1420)D; Q(1429)R; S(1471)N; S(1489)L; E(1504)D; S(1575)N; P(1582)S; K(1585)R; S(1588)P; N(1594)D; D(1606)N;	L(157)I; S(385)N; S(431)P; S(448)G; S(602)A; I(629)V; Q(692)R; A(769)S; I(1066)V; S(1237)N; E(1249)D; D(1252)E; R(556)G; V(1264)I; N(1306)K; R(556)G; S(1316)I; H(1333)Y; R(556)G; S(1351)T; A(1381)I; G(1417)T; N(1420)D; E(1423)D; I(1442)V; T(1465)V; R(1468)S; S(1471)N; D(1475)K; T(1483)A; E(1486)G; V(1492)T; E(1498)D; I(1501)A; E(1504)N; I(1510)L; K(1513)G; T(1516)V; K(1570)R; S(1576)N; D(1579)E; P(1583)S; N(1494)D; A(1600)T; D(1606)N; T(1609)A; N(1612)D; N(1541)S;

IUPAC code: A = Alanine, B = Aspartic acid, C = Cysteine, D = Aspartic Acid, E = Glutamic Acid, F = Phenylalanine, G = Glycine, H = Histidine, I = Isoleucine, K = Lysine, L = Leucine, M = Methionine, N = Asparagine, P = Proline, Q = Glutamine, R = Arginine, S = Serine, T = Threonine, V = Valine, and W = Tryptophan.

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
