# Peer review of "Isolation of a Divergent Strain of Bovine Parainfluenza Virus Type 3 (BPIV3) Infecting Cattle in China"

_viruses, 2019, doi:10.3390/v11060489_

Reviewer 1 Report

In this manuscript, Leal and colleagues isolated a new BPIV3 strain in China. Characterization of this virus has been performed adequately. Data are largely consistent with the interpretation. However, the contribution of this study to the scientific field regarding pathogenic viruses in cattle is not clear. The significance of this study should be illustrated in the text more clearly.

Major comments

1. The procedure how authors detect and isolate BPIV3 BJ is not clear in the present description. The example is in line 97-99. Did the authors detect BVDV, BPIV3, BHV-1 and BRSV by PCR?

2. Additionally, I suggest that the results of PCR for virus detection should be included in figures.

3. The CPE caused by BPIV3 infection is not clear in Figure 1. Is it possible to show a typical CPE by a magnified picture?

4. Lines 22-24 (Abstract), lines 53-55 (Introduction) and lines 144-146 (Discussion)

These sentences indicate the significance of this study. However, these descriptions seem ambiguous. Please rewrite them to express the importance of the present study concretely.

5. Line 113 “approximately 15480”

The genome length of paramyxovirus should be a multiple of 6 (rule of six). I think the word “approximately” is inappropriate.

Minor comment

1. Line 122 “figure 1” may be figure 2.

2. There are some grammatical errors in the manuscript. Please re-check and amend them.

Author Response

In this manuscript, Leal and colleagues isolated a new BPIV3 strain in China. Characterization of this virus has been performed adequately. Data are largely consistent with the interpretation. However, the contribution of this study to the scientific field regarding pathogenic viruses in cattle is not clear. The significance of this study should be illustrated in the text more clearly.

 Resp; Bovine parainfluenza-3 virus (BPIV3) is an endemic infection in cattle populations and underappreciated in many countries. Our study is important because is showing that there are many distinct strains of BPIV3 circulating in China. This diversity of BPIV3 lineages may impact on the preventive strategies and vaccine design. We also foundunique mutations in an essential immunodominant region (NP protein), demonstrating ongoing adaptation of BPIV3 for immune escape. The text was modified accordingly.

Major comments

1. The procedure how authors detect and isolate BPIV3 BJ is not clear in the present description. The example is in line 97-99. Did the authors detect BVDV, BPIV3, BHV-1 and BRSV by PCR?

 Resp; We did and results were included in the new version of the manuscript

2. Additionally, I suggest that the results of PCR for virus detection should be included in figures.

 Resp; We did include results of PCR assay aimed to detect other viruses

3. The CPE caused by BPIV3 infection is not clear in Figure 1. Is it possible to show a typical CPE by a magnified picture?

 Resp; We did change the figure 1 and increased the resolution

4. Lines 22-24 (Abstract), lines 53-55 (Introduction) and lines 144-146 (Discussion)

These sentences indicate the significance of this study. However, these descriptions seem ambiguous. Please rewrite them to express the importance of the present study concretely.

 Resp;

5. Line 113 “approximately 15480”

The genome length of paramyxovirus should be a multiple of 6 (rule of six). I think the word “approximately” is inappropriate.

  Resp; The word “approximately” was removed

Minor comment

1. Line 122 “figure 1” may be figure 2.

  Resp; We have corrected this in the manuscript.

2. There are some grammatical errors in the manuscript. Please re-check and amend them. Resp; the manuscript was reviewed by a professional grammar service

Reviewer 2 Report

Although the information presented in this paper is not really novel, it does contribute to the relatively small data base concerning the molecular epidemiology of bovine parainfluenza virus infection specifically as it relates to variants and genotype.  It therefore warrants publication; however more details regarding the primers and conditions of PCRs would aid others in being able to replicate these results and do further typing.  These details should be added before final acceptance. In the M and M reference is made to some of these details in "Table S1", but it is not apparent where those details actually are???

Author Response

Although the information presented in this paper is not really novel, it does contribute to the relatively small data base concerning the molecular epidemiology of bovine parainfluenza virus infection specifically as it relates to variants and genotype.  It therefore warrants publication; however more details regarding the primers and conditions of PCRs would aid others in being able to replicate these results and do further typing.  These details should be added before final acceptance. In the M and M reference is made to some of these details in "Table S1", but it is not apparent where those details actually are???
Resp; sequences of primers used and some details about the conditions of amplifications were included in the new version of the manuscript and the sequences of all primers are in the supplementary material.

Round  2

Reviewer 1 Report

The quality of this manuscript has been appropriately improved. However, I cannot find the Fig. 1 in this revised manuscript.

Author Response

We are sorry for the missing figure.

I have uploaded the manuscript with figure 1 and also a PDF file of this manuscript